# Development and Initial Testing of an Artificial Intelligence-Based Virtual Reality Companion for People Living with Dementia in Long-Term Care

**DOI:** 10.3390/jcm13185574

**Published:** 2024-09-20

**Authors:** Lisa Sheehy, Stéphane Bouchard, Anupriya Kakkar, Rama El Hakim, Justine Lhoest, Andrew Frank

**Affiliations:** 1Bruyère Research Institute, 43 Bruyère St., Ottawa, ON K1N 5C8, Canada; lsheehy@bruyere.org (L.S.); afrank@bruyere.org (A.F.); 2Cyberpsychology Lab of UQO, Université du Québec en Outaouais, Gatineau, QC J8X 3X7, Canada; 3HEC Liège—École de Gestion, Université de Liège, 4000 Liège, Belgium

**Keywords:** virtual reality, long-term care, dementia, artificial intelligence, large language models, compassion, reminiscence, elderly, cognitive decline, cognitive impairment

## Abstract

**Background/Objectives:** Feelings of loneliness are common in people living with dementia (PLWD) in long-term care (LTC). The goals of this study were to describe the development of a novel virtual companion for PLWD living in LTC and assess its feasibility and acceptability. **Methods**: The computer-generated virtual companion, presented using a head-mounted virtual reality display, was developed in two stages. In Stage 1, the virtual companion asked questions designed to encourage conversation and reminiscence. In Stage 2, more powerful artificial intelligence tools allowed the virtual companion to engage users in nuanced discussions on any topic. PLWD in LTC tested the application at each stage to assess feasibility and acceptability. **Results**: Ten PLWD living in LTC participated in Stage 1 (4 men and 6 women; average 82 years old) and Stage 2 (2 men and 8 women; average 87 years old). Session lengths ranged from 0:00 to 5:30 min in Stage 1 and 0:00 to 53:50 min in Stage 2. Speech recognition issues and a limited repertoire of questions limited acceptance in Stage 1. Enhanced conversational ability in Stage 2 led to intimate and meaningful conversations with many participants. Many users found the head-mounted display heavy. There were no complaints of simulator sickness. The virtual companion was best suited to PLWD who could engage in reciprocal conversation. After Stage 2, response latency was identified as an opportunity for improvement in future versions. **Conclusions**: Virtual reality and artificial intelligence can be used to create a virtual companion that is acceptable and enjoyable to some PLWD living in LTC. Ongoing innovations in hardware and software will allow future iterations to provide more natural conversational interaction and an enhanced social experience.

## 1. Introduction

Dementia is a collection of signs and symptoms that affect memory, executive function, language, and problem-solving to a degree which impacts daily life [1]. In Canada, the number of people living with dementia is expected to double from approximately 500,000 to over 900,000 between 2022 and 2030 [2]. Different but still dramatic estimates are in place for OECD countries (dementia increasing by 50% between 2021 and 2040, from 21 million to 32 million people), and for the entire world (increasing by 166% between 2019 and 2050, from 57.4 million to 152.8 million people) [3,4]. Twenty-five percent of adults over 85 years old have dementia [5], many of whom will require extensive support. As of 2016, in Canada, 69% of the residents living in long-term care (LTC) were living with dementia [6].

Feelings of social isolation are common in residents living in LTC due to factors such as communication difficulties, cognitive impairment, physical limitations, a lack of visitors, and low staffing levels [7]. Loneliness and isolation are associated with depression and reduced quality of life in older adults [8]. People living with dementia (PLWD) are at a much higher risk of depression, with depression presenting in 20–45% of those with dementia compared to 3.0–4.5% of all adults aged 65 and older [9]. Therefore, depression is very common in LTC residents (up to 44% of LTC residents in Canada), and is associated with deteriorating function, decreased quality of life, aggression, conflict with staff, and withdrawal [10].

Increasing LTC residents’ opportunities to interact socially contributes to improved mood and quality of life [11]. In one large study, person-centered care training for staff which promoted personalized activities and social interactions significantly improved the quality of life of LTC residents living with dementia [12].

Personalized activities, such as reminiscence therapy and storytelling, can be facilitated by photos, meaningful items (e.g., clothing or toys), music, or life stories (i.e., recorded oral histories), and can be used to improve communication, reinforce relationships between residents and caregivers, and strengthen family interactions [13]. Such therapy may reduce depression and improve the quality of life in LTC residents living with dementia [14]. Residents living with dementia can also interact with dolls, robotic pets, and social robots [15,16,17,18]. Robotic pets can respond to voice commands or touch while social robots can play simple video games, sing songs, or facilitate video calls. These innovations have been shown to improve communication, activity participation, mood, social interaction, and quality of life for older adults, residents living in LTC, and PLWD living in the community [15,16,17,18].

A novel approach to reducing the feelings of isolation and loneliness in PLWD living in LTC is through conversation with a computer-generated “virtual companion” presented while immersed in virtual reality (VR) using a head-mounted display. VR technology allows participants to become immersed in a computer-generated environment and interact via sensorimotor channels [19]. Furthermore, recent developments in large language model generative artificial intelligence (AI) now permit the interpretation of spoken inputs from a user, and the generation of emotionally appropriate responses [20]. By combining the technological potential of both VR (e.g., ref. [21]) and AI (e.g., ref. [22]), clinical benefits may be achieved.

PLWD living in LTC have been shown to react positively to VR technology [23,24]. Two studies of PLWD living in LTC reported no negative side-effects such as dizziness, nausea, or disorientation in VR sessions lasting up to an average of 8–10 min [25,26]. VR immersion in virtual nature, tourist locations, or relaxation scenarios has the potential to produce pleasure, well-being, improved mood, and alertness in PLWD, including those living in LTC [25,26,27,28]. LTC staff also believe that VR experiences can be beneficial to residents [29].

A virtual companion able to utilize aspects of immersive VR and AI may create an outlet for social interaction and conversation for PLWD living in LTC. In contrast to current AI applications dedicated to answering user queries, a virtual companion can actively engage the user to reminisce, potentially reducing loneliness and isolation, and improving quality of life [21]. Being immersed in VR and interacting with a conversational agent may also allow PLWD living in LTC to feel as though they are no longer in their LTC facility, a phenomenon called presence [30,31].

The goals of this study were to accomplish the following: (a) develop a novel virtual companion, presented using a head-mounted VR display, incorporating AI tools to allow conversational abilities, and (b) assess the feasibility and acceptability of this companion in PLWD living in LTC. Feasibility and acceptability assessments included the determination of the ease of use of the hardware and software for PLWD, the enjoyment and acceptability of the virtual companion, and the duration and quality of interactions between PLWD and the virtual companion [32].

## 2. Materials and Methods

The companion was developed in an iterative process followed by empirical data collection in a two-stage multi-method study design as defined by Elbanna [33], with each stage representing a different version of the virtual companion. At each stage, the development of the virtual companion is described, and the quantitative and subjective results of feasibility are reported to determine whether the intervention should be recommended for efficacy testing [32]. Research ethics approval was obtained from the Bruyère Research Ethics Board (Ottawa, ON, Canada; M16-21-032). The study conformed to the principles laid out in the Declaration of Helsinki and the Tri-Council Policy Statement: Ethical Conduct for Research Involving Humans in Canada.

### 2.1. Stage 1

#### 2.1.1. Hardware

The virtual companion was run on an Omen desktop gaming computer (Hewlett-Packard: Mississauga, Canada) and was presented using the Meta Quest 2 VR head-mounted display (Meta: Menlo Park, CA, USA) connected to the computer via a Quest Link cable (Meta: Menlo Park, CA, USA). The Meta Quest 2 is equipped with integrated audio microphone and speakers. The optional Elite Strap (Meta: Menlo Park, CA, USA) was available for the improved support of the head-mounted display. Users wore a disposable eye mask for hygienic purposes.

The head-mounted display and controllers were cleaned before and after each use with Accel INTERVention hospital-grade sanitizing wipes (Diversey: Fort Mill, SC, USA) and disinfected with a CX1 UVC LED decontamination unit (Cleanbox Technology: Nashville, TN, USA).

#### 2.1.2. Virtual Environment

The virtual environment and companion were developed by our team, which includes a cognitive neurologist with extensive experience assessing and treating PLWD, a clinical psychologist with extensive experience developing and deploying immersive VR for clinical applications, and a physiotherapist with a PhD in Rehabilitation Science having extensive clinical and research experience working with PLWD and older adults. More than ten clinicians and volunteers with experience working with PLWD, and a technical team with artistic and technical experience in VR and AI [34], were included. Every aspect of the development was created with the end-users (PLWD and their caregivers) in mind.

A virtual companion character, a young woman named “Kiera”, was presented sitting on a couch in a living room, with the user immersed in VR seated across the room (Figure 1). The appearance of the virtual companion was determined following discussions among the research team, artistic impressions, and social psychology studies suggesting that people are more likely to feel comfortable talking to and confiding in women [35].

The virtual environment was developed using 3D StudioMax version 2021 (Autodesk: San Francisco, CA, USA), Unity Store Assets version 2021 (Unity Technologies, San Francisco, CA, USA), and Character Creator version 3.4 (Reallusion: Atlanta, GA, USA). The animations of the virtual companion were created using Deepmotion version 2 (San Francisco, CA, USA), UMotions version 2021 (Soxware Interactive: Böhlerwerk, Austria), and Salsa version 2020 (Crazy Minnow Studio: Clear Lake, IA, USA). The virtual environment was rendered in stereoscopy using the Unity 3D engine version 2021 (Unity Technologies: San Francisco, CA, USA).

In Stage 1, the virtual companion interacted with the user using three AI modules: (a) speech-to-text (Recognissimo version 2021 from bluezzzy: San Francisco, CA, USA), (b) interpreting the intent of the text message and selecting pre-scripted replies (Neuralintents Library version 2021 from NeuralNine: Neuberg an der Mürz, Austria, and Flask Library version 1.1.1 in Python: Vienna, Austria), and (c) text-to-speech (using pre-scripted sentences with Speechelo 2021 from Videly Marketing: Dumbravita, Romania). The implemented solution required an operator to start the application, after which the virtual companion introduced herself and then spoke one of eight questions intended to encourage conversation and foster reminiscence, such as “Tell me something about your childhood”. After the user replied, the virtual companion could recognize 14 different intentions in the user’s sentence, and automatically reply with one of five responses inviting the user to continue the discussion, such as “Tell me more about that”. If the application did not detect a response, the operator could manually trigger the companion to provide her own anecdote in order to give the user topics about which to converse. The virtual companion was able to recognize two questions: “Where am I?” and “Who are you?” and automatically respond appropriately. The operator could manually repeat questions, and adjust the volume. At the end of the conversation, the virtual companion concluded with “It was a pleasure chatting with you today”. (See Lhoest, 2022 [34] for more technical details).

#### 2.1.3. Participants

PLWD living in two LTC residences in Ottawa, Canada were recruited. PLWD were eligible for inclusion if they were diagnosed with dementia of any cause and severity, were not on quarantine measures due to COVID-19, and were able to understand and speak English. Participants were excluded from the study if they had any contraindications to immersive VR (such as hypersensitivity to motion sickness, uncontrolled mood disorder, or seizure disorder), were medically unstable, actively lightheaded or vertiginous, had neck pain or headache, or had an active infection of the face or eyes. Free and informed consent was obtained from the participants’ substitute decision makers (the same person who would provide consent for medical decisions). While the participants were not deemed able to provide full consent for research, they could understand the general methods of the proposed study and what impact it may have on them. Interest in participating was obtained from the potential participants before contacting their substitute decision maker. Once consent was obtained, the participants provided verbal assent to the study at each contact point with the research team.

Demographic data were collected to describe the participant sample: age, gender, and Cognitive Performance Scale score indicating the level of dementia [scored from 0 (intact) to 6 (very severe impairment)] [36].

The LTC staff who worked directly with the participants provided free and informed consent to provide feedback on their observations of the participants’ interactions with the virtual companion.

#### 2.1.4. Feasibility and Acceptability Measures

The number of sessions per participant and the duration of each session with the virtual companion were recorded. The ease of use, comfort level, interaction and verbal communication with the virtual companion, apparent enjoyment, and interest to use the virtual companion again were collected from the PLWD, LTC staff, and the research team. At the end of each participant’s interaction with the virtual companion, the PLWD and LTC staff were asked questions in a semi-structured format (see Appendix A) to explore their interpretation of the experience. Suggestions for improvement were captured to aid in the ongoing iterative development of the virtual companion.

#### 2.1.5. Procedure

A research team member was trained to plug the head-mounted display into the computer, open the application, and run the virtual companion in the VR environment using the operator interface. The participants were seated comfortably and provided with a disposable eye mask. The head-mounted display was adjusted so that the participant was comfortable and could see the VR environment. The participants then conversed with the virtual companion with minimal interruption. Support was provided for the participants after their experience if they expressed distress.

#### 2.1.6. Subjective Analysis

The analysis of comments and semi-structured interviews were performed using a method similar to the framework coding method combining inductive and deductive components [37] using Excel 2019 spreadsheets. The coding and subsequent categorization included the following initial categories: hardware and software set-up, user experience, ease of conversation, and technology comments. As new feedback was collected, information was coded and previous codes and categories were adjusted. Disputes were discussed with a third researcher until consensus was obtained, and thematic relationships were established to analyze the data.

Demographic and user data were presented for each participant. Subjective data were categorized and summarized relative to each feasibility objective.

### 2.2. Stage 2

#### 2.2.1. Hardware

The hardware in Stage 2 was identical to that used in Stage 1. The disposable eye masks were cumbersome for PLWD using the head-mounted display, so their use was discontinued.

#### 2.2.2. Virtual Environment

The appearance of the virtual environment and companion remained the same as in Stage 1, except for small improvements to the animations of facial and non-verbal expressions, making the virtual companion appear friendlier. For example, the virtual companion smiled more and leaned forward at times to suggest active listening and interest in the conversation.

Significant modifications were included given the development of more powerful AI solutions for all three integrated AI modules. The speech-to-text module utilized Whisper version 2 (OpenAI: San Francisco, CA, USA). Importantly, the interpretation of the user’s intent and generation of replies was performed by the AI Generative Pre-trained Transformer (GPT) 3.5 large language model application programming interface (API) (OpenAI: San Francisco, CA, USA). Within the GPT 3.5 API settings, the companion’s persona was defined as a friendly nurse working with PLWD, who asks questions specifically to inspire reminiscence. Settings were created to automatically detect when the user stopped talking (i.e., a combination of minimal volume threshold to define a silence and duration of the silence), thereby removing the need for an operator to manually trigger responses. Text-to-speech of responses generated by GPT 3.5 was performed by ElevenLabs version 1 (ElevenLabs: New York, NY, USA) using a voice model selected from their library.

#### 2.2.3. Participants

In Stage 2, LTC residents living in one LTC residence in Ottawa, Canada were recruited. Only residents with mild cognitive impairment or mild-to-moderate dementia were considered, as they were more able to carry on a fluent conversation. LTC staff members were not recruited in Stage 2. The same procedures as in Stage 1 were applied in obtaining free and informed consent.

#### 2.2.4. Feasibility and Acceptability Measures

The outcome measures were identical to those used in Stage 1, with the exception that LTC staff members were not included.

#### 2.2.5. Procedure

The procedure was identical to that used in Stage 1. If participants were willing and available, they were invited to engage with the virtual companion a second and third time. Up to three sessions with the virtual companion were offered to each participant, spread out over several weeks.

#### 2.2.6. Subjective Analysis

The analysis was identical to that used in Stage 1.

## 3. Results

### 3.1. Stage 1

Ten PLWD living in LTC participated in Stage 1. The descriptive data and interactions with the virtual companion are reported in Table 1. The time spent interacting with the virtual companion ranged from 0:00 to 5:30 min, with an average of 2:48 min.

#### 3.1.1. Hardware and Software

Setting up the computer, monitor, cables, and head-mounted display was reported to be fairly simple for the trained research staff, and took less than 15 min. Six out of the ten PLWD and most LTC staff felt that the head-mounted display was comfortable, easy to wear, and well-fitted. However, for two out of the ten PLWD, the head-mounted display felt heavy, and for another two out of the ten, the head-mounted display shifted over their eyes easily, obscuring their view of the virtual environment. The Elite Strap was preferred by most users as it provided more support and allowed the head-mounted display to remain in place. The disposable eye masks frequently shifted and were difficult to place accurately over the eyes.

Several software challenges were experienced when setting up the Stage 1 version of the virtual companion. For example, the room with the virtual companion was sometimes seen to be tilted. This could be corrected with the operator interface. The sessions were brief given the limited options provided by the response generation module, which limited reciprocal conversation and deep exploration of any topic.

Prior to the conversation, there were occasions in which the virtual companion would not emit sound and had to be reloaded, or the software unexpectedly reverted to the loading screen. Frequently, the virtual companion experienced speech recognition issues due to a combination of software limitations and different patterns of speech often used by PLWD. The participants often mumbled, spoke few words at a time, or spoke in a quiet voice. The limited ability of the system to recognize speech or when the user was finished talking created both long pauses (if the participant spoke only a short phrase) and interruptions (if the participant talked more at length).

#### 3.1.2. Feasibility and Acceptability, and Subjective Reports

The conversation with the virtual companion flowed relatively naturally for some, and many PLWD, especially those with milder dementia, engaged well. Participant 2022-9 commented that the virtual companion was “friendly, (and) questions were good”. The LTC staff also observed that the virtual companion “is a friendly companion and asks interesting questions. Her few responses are well-done, and the anecdotes she gives are just the right length. Conversing with her is pleasant”. Indeed, according to both PLWD and LTC staff, the personal anecdotes provided by the virtual companion were helpful to the overall interaction, inducing reminiscence, inviting positive memories, and increasing engagement. Another frequent comment was that the length of the conversation was too short, and that pauses between the questions were not sufficiently long enough to provide a response. The limited number of questions asked by the virtual companion limited reciprocal conversation. One PLWD was confused by the virtual companion and four PLWD did not converse at all. The reasons included an unwillingness to try on the head-mounted display, attempts to remove it as it was being positioned, or being unable or unwilling to speak to the virtual companion.

Because of the severity of their dementia, not all PLWD were able to answer the interview questions regarding their experience with the virtual companion. Participants 2022-4 and 2022-5 mentioned that it was “nice to speak to someone”. A majority of the observed interactions involved good memories and social connection, and the LTC staff observed one participant engaging in what they felt was “a good chat”. Three PLWD specifically noted that visiting with the virtual companion improved their day and that they would interact with her again.

Further, an LTC staff noted that the virtual companion “is a friendly companion and asks interesting questions. She has a good tone and it is not difficult to suspend disbelief when speaking to her. She imitates a human’s behavior to a reasonable degree and the environment is welcoming”. However, another LTC staff mentioned that the virtual companion “seems somewhat realistic but lacks nonverbal cues, and can sometimes come off as blunt and robotic”.

There were no complaints of simulator sickness in Stage 1.

### 3.2. Stage 2

Ten new LTC residents living with cognitive impairment participated in Stage 2. The descriptive data and interactions with the virtual companion are reported in Table 2. The time spent interacting with the virtual companion ranged from 00:32 to 29:00 min for the first session (average 11:42 min), from 0:00 to 34:48 min for the second session (average 18:07 min), and from 1:07 to 53:50 min for the third session (average 29:39 min).

#### 3.2.1. Hardware and Software

Nine out of the ten participants found that the head-mounted display was heavy, especially after a long time. However, six out of the ten also reported that it was “comfortable”. For the participants with a reduced ability to hold their heads up, the Elite Strap was used for improved support. The head-mounted display did not fit well over eyeglasses.

Software challenges, such as error codes and freezing, tended to occur 25–30 min into the conversation. At times, the virtual companion unexpectedly started to speak words that were not English. At other times, the virtual companion interrupted the participants while they were talking. However, this did not necessarily create a negative experience according to the participants. The participants who had quiet and raspy voices, or spoke using single words or short phrases, were not able to trigger the microphone threshold to activate the listening mode of the AI speech-to-text module. When the participants provided long answers, the software processing time created long pauses in the conversation.

#### 3.2.2. Feasibility and Acceptability, and Subjective Reports

Half of the participants enjoyed conversing with the Stage 2 version of the virtual companion and provided positive comments. Participant 2023-3 mentioned that “She really knows her stuff”, and Participant 2023-2 observed that the virtual companion asked “… great questions, they seem to be written by a human and the questions were very stimulating and made me think”. On the other hand, two participants found the interactions overwhelming. Participant 2023-9 stated that “She doesn’t understand me and I don’t understand her”. Participant 2023-4 did not appreciate the virtual companion’s many questions, stating that “Yes, I liked, but too fast, can’t think, too many things”.

The conversation topics varied widely. Many intimate and mentally stimulating conversations took place, such as reminiscing about loved ones, sharing personal experiences and interests, and recalling recent interactions with grandchildren. In many instances, the questions generated by the virtual companion showed great emotional intelligence. However, in one case, the participant was discussing her husband’s death, and the virtual companion completely dismissed this, which seemed uncaring. While the conversations occasionally switched suddenly to other topics, most participants did not seem to notice. At the end of their session, one participant stated “Thank you so much, I usually don’t talk this much”.

There were varied opinions on the appearance and tone of the virtual companion. When asked if they liked the virtual companion and thought that they could trust her, Participant 2023-2 commented that the virtual companion appeared “too fake”, “artificial”, and “saw it was a robot”. Participant 2023-1 stated that they “thought she was a real person at first and then realized she wasn’t”. However, according to Participant 2023-3, “She talks the way I feel. I like her. She is very friendly. It is like I have known her for years. I like that we feel the same”.

Participant 2024-2 mentioned that the virtual companion’s hand motions and facial expressions were “stiff” and “she doesn’t imitate people”. On the other hand, many participants found the companion to be very knowledgeable and interesting. Participant 2023-4 “liked her as a person to do business with, very nice, beautiful and very professional responses … good at expressing words”. Participant 2023-3 commented that “she sounds trustworthy to me. She tells a story like she has been there, makes me feel close to her”.

Some participants felt that the experience improved their mood for the moment, though some did not find much value in it. Participant 2023-2 stated “it was fun, made me rethink. I would rather a real person, but it is better than nothing in an area with less people”. Participant 2023-6 stated how the companion “… made me happy to see there is a person like this around”, while Participant 2023-3 stated “I feel very comfortable, not shy”. On the other hand, Participant 2023-7 felt “I think that she is alright, (but) I did not see any point in the conversation”. Six out of ten participants agreed that they would be interested in spending time with the virtual companion again.

There were no complaints of simulator sickness in Stage 2.

## 4. Discussion

This project documents the iterative development of an AI-based “virtual companion”, designed for PLWD, through two stages of development. Stage 1, while using a rudimentary version of the companion and AI tools, showed that some PLWD could tolerate wearing a VR head-mounted display and interact with the virtual companion for a few minutes. The software was limited in the questions the companion could pose, and there were delays and some difficulties with voice recognition. It was also recognized that not all PLWD would accept wearing the head-mounted display, or speaking to a stranger about personal topics. Nevertheless, the results supported proceeding to Stage 2.

Conversing with a virtual companion was more feasible for people with milder dementia who were much more verbal and likely to reminisce and tell stories. For this reason, the inclusion criteria were revised in Stage 2 to focus on participants with mild cognitive impairment or mild-to-moderate dementia.

During Stage 2, the virtual companion benefited from recent (as of the autumn of 2023) advances in AI technology to converse much more effectively on a broader range of topics. n particular, generative AI was used for the response module, and an AI solution able to generate natural speech in real time was employed for the text-to-speech module. Software limitations continued to cause conversation delays and interruptions. Because AI systems do not comprehend what the user is saying (for more on how large language models operate and reproduce human-like interactions, see [38]), it is difficult for AI systems to determine exactly when the user is finished expressing their thought. Indeed, a moment of silence when the user is speaking may represent a pause in the ongoing expression of an idea, mumbling, stuttering, a raspy voice, or in fact that the user is finished talking. The available speech-to-text module automatically processed audio information in one batch when the user was considered to be done talking, sometimes leading to noticeable delays while the system was processing all the information at once. At other times, there were interruptions by the virtual companion when users were still talking. However, the substantial increase in conversation length and positive feedback from several participants provided support for yet further development.

VR head-mounted display units are becoming more commonplace, and have been generally well tolerated by residents living in LTC, as well as their care staff [23,24,29]. In our study, a consistent complaint was that the headset was heavy. While the size and weight of VR head-mounted displays may decrease in the future, weight should be considered when choosing a particular display unit for use in this population. Some participants complained of itchiness and irritation, potentially caused by the silicone portion of the headset that contacted the skin, or the cleanser used to sanitize the unit. Furthermore, the head-mounted display did not fit comfortably over many styles and sizes of eyeglasses. Future versions of the virtual companion should be compatible with a variety of different head-mounted displays.

As noted, the availability of generative AI solutions enabled the virtual companion to converse in detail on topics of the user’s choosing, for up to almost one hour. The use of more advanced AI technology will help to remove some of the conversation delays and interruptions, which will make interactions with the virtual companion smoother and more realistic. Tailoring the generative AI system to PLWD was very helpful, resulting in a seemingly emotionally intelligent companion able to inspire reminiscence. The focus on engaging the user to reminisce sets this application of AI apart from common applications which focus on decision-making support, assisting therapists, and counselling [39].

Findings from Stage 2 have led to yet further improvements to the virtual companion. Recent (as of the summer of 2024) AI solutions integrating speech, text, and response generation can circumvent many interruptions and delays associated with defining when the user has stopped talking. A new user interface allows choice among the different large language models (from GPT 3.5 to GPT 4o), AI text-to-speech models, and AI voices provided by OpenAI. Animations while the virtual companion is speaking have been improved. The experience can now take place in a virtual apartment or a virtual park (see Figure 2) providing more options to induce a positive mood [40] in people living in LTC.

The virtual companion was presented in an immersive VR environment using a head-mounted display. The advantage of immersive VR lies in the feeling of presence [30,31], which is the feeling of being somewhere else, rather than, for example, an LTC residence (i.e., the place illusion); the feeling that what is happening in VR is real (i.e., the plausibility illusion); and the feeling of engaging with a real person (i.e., social presence). Some participants made comments suggesting that they indeed experienced presence. Presence has been found to play a significant role in perceived social support [41], the motivation to use conversational agents in human–AI interactions [42], and anthropomorphism, the attribution of human characteristics to a non-human object [43]. The use of immersive VR platforms has led to improved cognitive and motor rehabilitation in PLWD [44]. One recent study [45] determined that the use of immersion during cognitive training provided greater benefits for global cognition than more conventional methods (such as paper-and-pencil or computer tasks) in people living with mild cognitive impairment. However, they found that lower levels of immersion provided more benefit than full immersion, possibly because older adults are less comfortable with full immersion technologies. Therefore, further study is necessary to investigate the role of immersiveness. Importantly, there were no complaints of simulator sickness in our PLWD participant sample, supporting the assertion that immersive VR is well tolerated by PLWD [25,26].

We used an animated computer-generated character instead of a photo-realistic image for the visual representation of the virtual companion. Complex real-time social interactions with photo-realistic images or 360-degree video recordings represent a technical challenge [46]. The impact of small imperfections in a photo-realistic image or in animations can create a sense of “eeriness”, while a less life-like image may be more relatable. This tendency has been shown to be even greater in immersive VR [47,48]. Feedback from some participants was that the companion appeared blunt, robotic, stiff, or fake. However, others described her as friendly. Future iterations of the virtual companion will improve the animations of facial expressions and non-verbal behavior. Further studies should assess the impact of virtual companion characteristics (e.g., gender, ethnicity, appearance, and voice) and virtual contexts (e.g., locations relevant to users’ life history). The impact of imperfections in verbal and non-verbal behaviors and interruptions must be assessed. Contrary to expectation, a lack of realism has been shown to be trivial among people with anxiety disorders [49] or substance use disorders [50] when the right stimuli were presented in the virtual environment to trigger emotional reactions.

Encouraging PLWD to engage with a virtual companion raises ethical considerations. PLWD may believe that the virtual companion is an actual person, and some participants did mention this. The use of generative AI in Stage 2 of our study increasingly produced human-like conversation, compounding this effect. It has been argued that a robot (or perhaps an AI conversational agent) that acts as if it cares for others can connect with older adults as well or better than human beings who may avoid emotional attachment [51]. So long as the use of a virtual companion does not negatively impact care by healthcare providers, and does not cause distress or become excessive, the benefits of a pleasant recreational activity meet the principles of respect for persons, beneficence, non-maleficence, and justice (for more on the ethics of using chatbots and conversational agents see [52,53]). Large language models have been known to be occasionally inaccurate or produce false claims that have been anthropomorphized by the term hallucination [38]. Although false statements or factual errors generated by AI may pose a risk to vulnerable populations, this is unlikely in an application which is designed to be empathetic and favor reminiscence. The virtual companion is designed to avoid engaging in discussions on delicate topics or providing advice, and to ask questions to the user instead of generating new ideas or explanations about factual events. Our application promotes and supports *reminiscence of a user’s own personal history*, which limits the potential risk for people with mental or cognitive disorders of being exposed to inaccuracies, disinformation, or dangerous advice (for more on this topic, see [54]). Future applications should use AI models that are more interpretable and transparent in their internal processing, reasoning, and decision-making patterns than mainstream large language models, and they should use generative AI models specifically developed for empathy and the detection of emotional reactions (e.g., refs. [55,56,57]).

One limitation of this study is that the feasibility of caregivers or LTC staff directly operating the virtual companion was not tested. This will be assessed in future studies to determine if staff or other caregivers feel that using the virtual companion is both feasible and worthwhile. Reminiscence therapy [14,27] and virtual social interactions [17,18] can be quite beneficial for people living in LTC; therefore, testing the impact of AI-based VR companions on mood, quality of life, and responsive behaviors in PLWD via randomized controlled trials is necessary.

## 5. Conclusions

A virtual companion was created using multiple AI-based components to rapidly transform speech into text, generate complex and human-like responses, and rapidly engage in conversations using a naturalistic human voice. The virtual companion was presented to PLWD using a VR head-mounted display providing a virtual environment that differed significantly from their LTC facility. Twenty PLWD living in LTC interacted with the virtual companion, some through meaningful and prolonged conversations, providing proof-of-concept that this technology may be beneficial in this population. Ongoing innovations in hardware and software will allow virtual companions to provide more natural conversational abilities and an enhanced social experience.

## Figures and Tables

**Figure 1 jcm-13-05574-f001:**
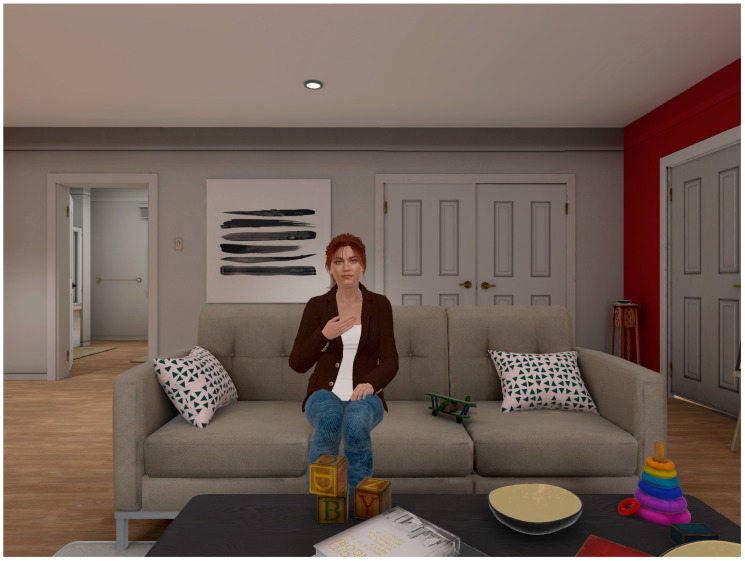
Screenshot of the virtual companion, “Kiera”, as seen by the person immersed in virtual reality.

**Figure 2 jcm-13-05574-f002:**
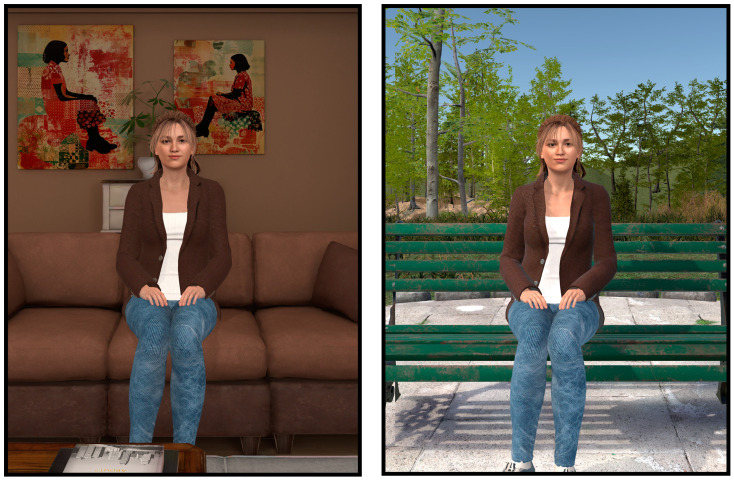
Screenshots of the improved virtual environments for the virtual companion in an apartment (**left**) or in a park (**right**).

**Table 1 jcm-13-05574-t001:** Descriptive data and virtual companion interaction data for the Stage 1 research sample. All the participants participated in one session with the virtual companion. M—man; W—woman.

Participant Number	Gender	Age	Cognitive Performance Score	Length of Session (min:s)	Subjective Reports
2022-1	M	68	3	3:00	The participant had a natural conversation with reminiscence on topics they had not thought about for a long time.
2022-2	W	85	3	1:00	The participant did not speak to the virtual companion.
2022-3	M	81	3	0:00	The participant did not want to wear the head-mounted display.
2022-4	W	93	4	4:30	The participant engaged in some conversation, but used semi-coherent sentences. There was minimal reminiscing.
2022-5	W	74	5	2:00	There was minimal interaction. The participant was happy to be asked questions, and responded with a smile.
2022-6	M	85	3	3:30	The participant engaged with the virtual companion and seemed pleased to answer the questions.
2022-7	W	82	5	4:00	The participant did not interact verbally, but was happy to wear the head-mounted display.
2022-8	W	80	3	2:30	The participant did not seem to understand the virtual companion, and was not interested in interacting.
2022-9	M	84	3	2:00	There was a short interaction. The participant expressed enjoyment and thought that the virtual companion was “friendly”.
2022-10	W	87	5	5:30	The participant responded to some questions but not others. There was some laughter.
Frequency or mean (standard deviation)	4 M 6 W	82 (6.9)	3.7 (0.9)	2:48 (1:42)	

**Table 2 jcm-13-05574-t002:** Descriptive data and virtual companion interaction data for the Stage 2 research sample. M—man; W—woman.

Participant Number	Gender	Age	Cognitive Performance Score	Number of Sessions	Length of Session(s) (min:s)	Subjective Reports
2023-1	W	83	1	3	7:32 3:39 2:07	Session 1—The participant’s voice was mistaken by the software for a language other than English. Session 2—There was an engaging conversation about the participant’s children. The participant enjoyed how immersive the conversation was. Session 3 –The participant gave one-word answers, so the virtual companion kept asking new questions.
2023-2	W	87	1	3	25:00 25:00 53:50	The participant had stimulating, deep, and personal conversations about traveling, teaching, poetry, and death. The participant did not believe that the virtual companion was a real person.
2023-3	W	87	3	3	20:23 29:00 32:53	The participant was very engaged, and did not understand that the virtual companion was not a real person. The participant had deep and intimate conversations about their family.
2023-4	M	93	3	3	22:39 33:53 30:46	Session 1—The participant was engaged but the conversation was “all over the place”. Session 2—The participant was very engaged, and talked about their farm. Session 3—This was primarily a question-and-answer conversation (with questions from the virtual companion), but the participant was engaged.
2023-5	W	92	3	1	0:32	The participant asked to take the head-mounted display off as they found it too heavy.
2023-6	W	93	1	2	29:00 34:48	The participant had very positive, lengthy conversations about their pet (Session 1) and cuisine/food (Session 2).
2023-7	W	96	2	2	2:23 9:10	Session 1—The conversation was dry, and the participant thought that the questions were “stupid”. Session 2—The participant gave one-word answers so the software was not able to engage and respond appropriately.
2023-8	M	86	4	1	5:50	The participant answered questions but felt overwhelmed.
2023-9	W	86	2	2	0:52 0:00	Session 1—The participant had a hard time understanding the virtual companion but was willing to try a second time. Session 2—The participant immediately asked to remove the headset.
2023-10	W	67	3	2	2:49 9:29	Session 1—The participant took the head-mounted display off but was willing to try a second time. Session 2—The software had difficulty understanding the participant’s speech (due to stuttering) but the participant appeared to enjoy the experience.
Frequency or mean (standard deviation)	2 M 8 W	87 (8.1)	2.3 (1.1)	2.2 (0.8)	Session 1: 11:42 (11:13) Session 2: 18:07 (14:04) Session 3: 29:54 (21:15)	

## Data Availability

The dataset is not publicly available due to privacy and ethical restrictions. The data for this study is available upon request addressed directly to the Research Ethics Boards of the Bruyère Research Institute (REB@bruyere.org), which must first approve the request. If the request is approved, anonymized data supporting the conclusions of this manuscript will be made available by the corresponding author.

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
