# Peer review of "Development and Initial Testing of an Artificial Intelligence-Based Virtual Reality Companion for People Living with Dementia in Long-Term Care"

_jcm, 2024, doi:10.3390/jcm13185574_

Round 1
Reviewer 1 Report
Comments and Suggestions for Authors
Summary
In the study entitled ‘Development and Initial Testing of an Artificial Intelligence-based Virtual Reality Companion for People Living with Dementia Living in Long-Term Care’ a two stage feasibility trial is described, in which Canadian residents of long term care facilities living with dementia are provided an opportunity to converse with an AI chatbot presented within an immersive VR environment facilitated by headset. Such conversation, it is argued, has benefits to people living with dementia. Stage one findings indicated model acceptability to be limited by more severe dementia, restricted question repertoire, and speech recognition flaws. Stage two findings indicate that improvements to the model, in particular the use of generative AI, together with exclusion of individuals exhibiting most severe dementia, yielded sufficient positive acceptability data to warrant further development of this technology.
Thank you for the opportunity to review this interesting application of recent technological innovations to improve the lives of people living with dementia. The novelty and timeliness of the application will be of interest to the clinical and research community. It is this reviewer’s opinion that the study is appropriate after minor revisions based upon the following points.
Abstract
Appropriate and well written in adherence with Journal of Clinical Medicine guidelines.
Introduction
Is concise and well evidenced. Study rationale is provided and supported by recent empirical findings.
Review Points:
1. Provide brief description of dementia to introduce the key study population.
2. Ng et al. (2024) citation not present in bibliographic references.
Methods
Ethics declarations at international and local level provided. Reference to ethics board approval provided. Overarching framework specified. Details of each hardware and software component are adequately specified to enable replication as far as feasible.
Review Points:
3. There is no rationale provided for the design choices made regarding salient aspects of the virtual companion’s presentation, such as sex, ethnicity, voice, dress. A rationale based upon empirical literature should be provided.
4. Page 4 line 151: ‘intensions’ ought to be ‘intentions’ in the context of intent recognition
5. Consent provided by substitute decision makers does not necessarily fulfil informed consent criteria – vulnerable population may have adverse experience despite the reported assent. Although the complexity of this process in Canada has recently been highlighted, recommendations have been made to forefront the individual in the consent process https://doi.org/10.1017/S0714980824000217 (see also https://doi.org/10.1080/13607863.2023.2264216 for discussion of consent vs. assent). More information regarding the consent process should be provided to allow the reader to assess the extent to which the research adhered to such recommendations. Notably, debrief is not clearly documented; it is important to be able to verify that the participants were given support following the intervention.
6. Pg 5 Line 190: Review grammar “was comfortable could see the VR environment”
7. Pg 5 Line 211: Please specify details of how this was achieved: “make the virtual companion appear friendlier.”
8. Pg 5 Line 225: The modification of stage 2 inclusion criteria, as is captured in Pg 12 lines 378-379, should be made explicit here also.
Results
Excellent level of descriptive, feasibility and subjective reports are provided. Tables provide concise information that is all relevant and aids the reader to interpret the findings.
Discussion
Need for more rigour in measurement for acceptability in future extensions to this feasibility study are mentioned. Discussion around ethical considerations are very welcome to see.
Review Points:
9. Pg 13 line 421: Review grammar “and allows implement strategies to circumvent the problems of interruptions”
10. The options for different virtual environments, described in Figure 2, are welcome. It would enhance this section to include a reference to work on VR environments and mood.
11. Use of LLMs such as ChatGPT introduce inherent biases and flaws such as hallucination that will present risks to the vulnerable population that are difficult to control for. How development of future applications will account for this should be mentioned.
Comments on the Quality of English LanguageQuality of English Language is high throughout. Minor grammatical suggestions have been made in the attached review file.
Author Response
It is this reviewer’s opinion that the study is appropriate after minor revisions based upon the following points.
We thank the reviewer for the very thoughtful and positive comments. We have addressed all comments and hope you will find it satisfactory.
Review Points:
- Provide brief description of dementia to introduce the key study population.
We added the following brief description from the Alzheimer’s Association (lines 35-37): Dementia is a collection of signs and symptoms that affect memory, executive function, language and problem-solving, to a degree which impacts daily life.
- Ng et al. (2024) citation not present in bibliographic references.
This unfortunate mistake has been corrected. This important reference is now present in the Reference list.
- There is no rationale provided for the design choices made regarding salient aspects of the virtual companion’s presentation, such as sex, ethnicity, voice, dress. A rationale based upon empirical literature should be provided.
We would have preferred to offer several options for the virtual companion (e.g., gender, ethnicity). However, due to budget limitations, we had to begin with only virtual character. The decision was not made following an elaborate systematic analysis or focus group discussions. The scientific literature was briefly consulted and the team discussed several options based on clinical impressions with our population and input from the artistic team. The revised manuscript now includes the rational and a reference from a meta-analysis consulted at the time (lines 136-139): The appearance of the virtual companion was designed following discussions among the research team, artistic impressions, and social psychology studies suggesting that people are more likely to feel comfortable talking to and confiding in women (Dindia & Allen, 1992).
- Page 4 line 151: ‘intensions’ ought to be ‘intentions’ in the context of intent recognition.
Thank you for pointing out this typo. It is now corrected (line 158).
- Consent provided by substitute decision makers does not necessarily fulfil informed consent criteria. Although the complexity of this process in Canada has recently been highlighted, recommendations have been made to forefront the individual in the consent. More information regarding the consent process should be provided to allow the reader to assess the extent to which the research adhered to such recommendations. Notably, debrief is not clearly documented; it is important to be able to verify that the participants were given support following the intervention.
To ensure we obtained consent by both the participants and the legal entity that must provide consent for them, we proceeded in three steps. First, we asked participants if they were interested. If they confirmed their interest, then the substitute decision maker was contacted and had to provide free and informed consent. We now mention in the article this is the same person who needs to consent for medical procedures. When consent was provided by the legally appointed substitute decision maker, participants were then approached and had to provide free and informed consent at each contact with the research team. This is referred as assent from participants, because this is the legally correct term in Canada to refer to this form of consent. As for support following the intervention, the manuscript now mentions it was offered to participants if needed. Note that no participant required such support.
The revised manuscript now states (Lines 175 – 181) that free and informed consent was obtained from participants’ substitute decision makers (the same person who would provide consent for medical decisions). While participants were not deemed able to provide full consent for research, they could understand the general methods of the proposed study and what impact it may have on them. Interest in participating was obtained from potential participants before contacting their substitute decision maker. Once consent was obtained, participants were reminded of the study and provided verbal assent at the time of each contact with the research team.
And on lines 203 – 204: Support was provided for participants after their experience if they were visibly distressed or verbally expressed distress.
- Pg 5 Line 190: Review grammar “was comfortable could see the VR environment”
The phrase was revised to add “and” (line 203).
- Pg 5 Line 211: Please specify details of how this was achieved: “make the virtual companion appear friendlier.”
Additional details are now provided on lines 225 to 227: For example, the virtual companion smiled more and leaned forwards at times to suggest active listening and interest in the conversation.
- Pg 5 Line 225: The modification of stage 2 inclusion criteria, as is captured in Pg 12 lines 378-379, should be made explicit here also.
As recommended, the information is now also mentioned in lines 241 to 244: Only residents living with mild cognitive impairment or mild-to-moderate dementia were considered, as they were more able to carry on a fluent conversation.
- Pg 13 line 421: Review grammar “and allows implement strategies to circumvent the problems of interruptions”
The text now reads as (lines 434-435): AI solutions have been chosen that integrate all three modules and allow the implementation of strategies to circumvent the problems of interruptions and delays associated with defining when the user is done talking.
- It would enhance this section to include a reference to work on VR environments and mood.
As requested, the manuscript now mentions (lines 441-442) a systematic review on the impact of the environment on mood. This recent review by Di Pompeo et al. includes studies on VR and without VR.
- Use of LLMs such as ChatGPT introduce inherent biases and flaws such as hallucination that will present risks to the vulnerable population that are difficult to control for. How development of future applications will account for this should be mentioned.
We thank the reviewer for giving us the opportunity the share our ideas about LLMs flaws. We addressed this in the paragraph discussing ethical considerations. In essence, the manuscript now mentions LLMs hallucinations, risks for people with mental or cognitive disorders, why it does not appear to be a significant source of concern in the context of the current study, and what future applications should do to account for the problem and circumvent it.
The following text has been added to the manuscript (lines 493-508): Large Language Models have been known to sometime be inaccurate or produce false claims that have been anthropomorphized by the term hallucination (Hicks et al., 2024). Although false statements or factual errors generated by AI may pose a risk to vulnerable populations, this is unlikely in applications such as ours designed essentially to be empathetic and favor reminiscence. The virtual companion is designed to avoid engaging in discussions on delicate topics or provide advice, and to ask questions of the user instead of generating new ideas or explanations about factual events. Users of the application are looking for discussions that promote and support reminiscence of their own personal history, which limits the potential risks for people with mental or cognitive disorders of being exposed to inaccuracies, disinformation or dangerous advice (for more on this topic, see Monteith et al., 2024). Ideally, future applications should use AI models that more interpretable and transparent in their internal processing, reasoning, and decision-making patterns than mainstream Large Language Models, and use generative AI models specifically developed for empathy and detection of emotional reactions in users by autonomous virtual agents (e.g., Joudeh et al., 2024; Gomez-Zaragoza et al., 2023; Parra et al., 2021).
Following the addition of this information in the manuscript, we added a brief mention about the possibility in our revision of the application to select LLMs (lines 436 to 438) and replaced the keyword Technology by Large Language Model.
Reviewer 2 Report
Comments and Suggestions for Authors
This an interesting paper that review, the use of a virtual companion in people with dementia living in Long Term Facilities in a pilot study. Minor changes should be done.
1. Please include in the introduction also data on the prevalence of dementia in the World and in developed OECD countries.
2. Line 94 in an academic paper the slash should be avoided "enjoyment/acceptability of the virtual companion". You could use "enjoyment and acceptability" if both were evaluated separately, or "enjoyment or acceptability" or rephrase the sentence.
3. In Material and Methods Please explain who is the substitute decision makers and how are they chosen in Canada, so a person not familiar can understand it.
4. Is this correct ? Informed Consent Statement: Informed consent was obtained from all participants’ substitute de-514 cision makers and also from all participants involved in the study. Reading the paper I had the impression, that when the patient couldn’t consent, the substitute maker ,should give the consent. Any way explain the process with more detail.
5. Line 151, I think it should be intentions, not intensions.
6. 2.1.6 Subjective Analysis Was any software used for coding? (atlasti o similar)
Author Response
Minor changes should be done.
We thank the reviewer for the very thoughtful and positive comments. We have addressed all comments and hope you will find it satisfactory.
- Please include in the introduction also data on the prevalence of dementia in the World and in developed OECD countries.
Following your recommendation, the following text was added (lines 39-42): Different, but still dramatic, estimates are in place for OECD countries (increasing by 50% between 2021 and 2040, from 21 million to 32 million people), and for the entire world (increasing by 166% between 2019 and 2050, from 57.4 million to 152.8 million people) (Nichols, 2022; OECD, 2023).
- Line 94 in an academic paper the slash should be avoided "enjoyment/acceptability of the virtual companion". You could use "enjoyment and acceptability" if both were evaluated separately, or "enjoyment or acceptability" or rephrase the sentence.
Thank you for pointing this out and suggesting a solution. The text has been revised accordingly on l;ine 99.
- In Material and Methods Please explain who is the substitute decision makers and how are they chosen in Canada, so a person not familiar can understand it.
A clarification on this topic was also requested by Reviewer 1. We now refer to the person who provides consent for medical decisions to explain who this person is without having to provide legal and technical details. The revised manuscript now states (Lines 175 – 181) that free and informed consent was obtained from participants’ substitute decision makers (the same person who would provide consent for medical decisions). While participants were not deemed able to provide full consent for research, they could understand the general methods of the proposed study and what impact it may have on them. Interest in participating was obtained from potential participants before contacting their substitute decision maker. Once consent was obtained, participants were reminded of the study and provided verbal assent at the time of each contact with the research team.
- Is this correct ? Informed Consent Statement: Informed consent was obtained from all participants’ substitute decision makers and also from all participants involved in the study. Reading the paper I had the impression, that when the patient couldn’t consent, the substitute maker, should give the consent. Any way explain the process with more detail.
This question was addressed in the same revision mention above in your point #3 (lines 175-181). To ensure we obtained consent by both the participants and the legal entity that must provide consent for them, we proceeded in three steps. First, we asked participants if they were interested. If they confirmed their interest, then the substitute decision maker was contacted and had to provide free and informed consent. We now mention in the article this is the same person who needs to consent for medical procedures. When consent was provided by the legally appointed substitute decision maker, participants were then approached and had to provide free and informed consent at each contact with the research team. This is referred as assent from participants, because this is the legally correct term in Canada to refer to this form of consent.
- Line 151, I think it should be intentions, not intensions.
Thank you for pointing out this typo. The correction was done on line 158.
- 2.1.6 Subjective Analysis Was any software used for coding? (atlasti o similar)
We did not us any specialized software for coding. We only use Excel, which is now mentioned in the revised manuscript (line 208).